# Molecular engineering of dihydroxyanthraquinone-based electrolytes for high-capacity aqueous organic redox flow batteries

Shiqiang Huang[1,3], Hang Zhang [1,3], Manohar Salla[1], Jiahao Zhuang[1], Yongfeng Zhi[1], Xun Wang[1] & Qing Wang [1,2] ✉

Aqueous organic redox flow batteries (AORFBs) are a promising technology for large-scale electricity energy storage to realize efficient utilization of intermittent renewable energy. In particular, organic molecules are a class of metal-free compounds that consist of earth-abundant elements with good synthetic tunability, electrochemical reversibility and reaction rates. However, the short cycle lifetime and low capacity of AORFBs act as stumbling blocks for their practical deployment. To circumvent these issues, here, we report molecular engineered dihydroxyanthraquinone (DHAQ)-based alkaline electrolytes. Via computational studies and operando measurements, we initially demonstrate the presence of a hydrogen bond-mediated degradation mechanism of DHAQ molecules during electrochemical reactions. Afterwards, we apply a molecular engineering strategy based on redox-active polymers to develop capacity-boosting composite electrolytes. Indeed, by coupling a 1,5-DHAQ/poly(anthraquinonyl sulfide)/carbon black anolyte and a $[Fe(CN)_6]^{3-/4-}$ alkaline catholyte, we report an AORFB capable of delivering a stable cell discharge capacity of about 573 mAh at 20 mA/cm$^2$ after 1100 h of cycling and an average cell discharge voltage of about 0.89 V at the same current density.

Significant progress has been made in exploiting renewable energy sources (i.e., solar and wind) in the past decades. However, their deployment in grid-connected power systems is limited by intermittency and volatility[1]. Electrochemical energy storage (EES) systems provide an important means to reversibly convert and store the produced electricity from solar and wind into chemical energy, which buffers the fluctuating power output of renewable generators and their impact to the power grids[2]. Among all the EES systems, aqueous redox flow batteries (ARFBs) because of the salient feature of decoupled power generation and energy storage, have great advantages of operation flexibility, scalability as well as safety, particularly suitable for large-scale application[3,4]. All-vanadium RFBs as the most mature

flow battery technology have been demonstrated at MW/MWh scale[5,6], while the high materials cost and potential environmental concern hinder their widespread deployment[7].

In recent years, a number of charge-storage materials based on organic redox-active molecules have been reported in aqueous organic redox flow batteries (AORFBs) because of their synthetic tunability, natural abundance, and inherent safety[8–11]. Quinone-based molecules have attracted considerable attention as they have tuneable redox potentials, good electrochemical reversibility and reaction rates in a broad pH range[12–17]. For instance, dihydroxyanthraquinone (DHAQ) derivatives generally have 2-electron reduction at relatively low potentials and thus have been explored

[1]Department of Materials Science and Engineering, National University of Singapore, Singapore, Singapore. [2]National University of Singapore (Suzhou) Research Institute, Suzhou, Jiangsu, PR China. [3]These authors contributed equally: Shiqiang Huang, Hang Zhang. ✉e-mail: msewq@nus.edu.sg

as anolyte for AORFBs[18], while the poor chemical stability and water solubility limit their application[12,17]. Although a good progress has been made by introducing functional groups to DHAQs to enhance the solubility and stability, the overall EES performance is still not satisfactory[13,14,19,20]. Among the reported DHAQ derivatives, 3,3′-(9,10-anthraquinone-diyl)bis(3-methylbutanoic acid) (DPivOHAQ) shows the lowest capacity fading rate but suffers from a low solubility of 0.74 M[19]. In addition, while 1,8-bis(2-(2-(2-hydroxyethoxy)ethoxy)-ethoxy)anthracene-9,10-dione (AQ-1,8-3E-OH) has a high solubility of 2.24 M, the chemical stability is poor[13]. More importantly, although the degradation mechanism of DHAQs has been proposed and the products are detected[21], there is still a lack of in-depth understanding of the mechanism which could provide further guidance to the electrolyte design.

Here the chemical stability of a series of commercial DHAQs was scrutinized both experimentally and computationally, and a hydrogen bonding-induced reaction mechanism was deciphered for the degradation of DHAQs, which reveals important implications for circumventing the stability issue and provide guidance to the design of stable redox molecules. Moreover, a molecular engineering strategy is proposed for fast selection of capacity-boosting materials to increase the volumetric capacity of AORFBs through single molecule redox targeting (SMRT) reactions[22–26]. The demonstrated 1,5-DHAQ/poly (anthraquinonyl sulfide) (PAQS)/carbon black electrolyte system has an effective solubility of 1,5-DHAQ as high as 2.6 M (5.2 M electrons), and exhibits appealing chemical stability with a low-capacity fading rate of 0.02% per day upon more than 1000 h′ test in an AORFB cell.

## Results

### Investigation on the hydrogen bond-mediated degradation mechanism of DHAQ molecules

The physicochemical properties of DHAQs are affected by the substituents on the aromatic rings[27]. To have a holistic understanding of the structure-property relation, a series of DHAQ isomers (Supplementary Fig. 1) with varying substitution sites of hydroxyl groups (i.e., 1,2-DHAQ, 1,4-DHAQ, 1,5-DHAQ, 1,8-DHAQ and 2,6-DHAQ) were investigated in terms of redox potential ($E_{1/2}$) and solubility (Supplementary Figs. 2–5 and Supplementary Table 1). Among all the DHAQs, although 2,6-DHAQ has the lowest $E_{1/2}$ (−0.706 V vs. SHE in 1 M KOH at $25 \pm 1$ °C) and the highest solubility (0.60 M in 1.5 M KOH at $25 \pm 1$ °C), it has been reported that the irreversible dimerization of anthrone intermediate during the reduction process severely impairs its stability[21]. However, the root cause for the side reaction remains unclear.

Here operando attenuated total reflection Fourier transform infrared spectroscopy (ATR-FTIR) was performed to scrutinize the changes of bonding structures by monitoring the reactions of 2,6-DHAQ at different stages of reduction. The operando ATR-FTIR measurement was conducted with a commercialized spectro-electrochemical cell (Supplementary Fig. 6) upon cyclic voltammetric (CV) scans. Figure 1a and Supplementary Figs. 7, 8 show the time-dependent transmittance difference (ΔT) of the FTIR spectra of 2,6-DHAQ during the reduction and oxidization processes in the first cycle of CV scan and the subsequent two scans, respectively. The vibrations of C−O and C=C bonds of 2,6-DHAQ in the range of 1300–1600 cm⁻¹ were recorded during CV scans. The signal of C=C gradually diminished in the cathodic scan and re-emerged in the anodic scan while C−O showed a reversed trend. Interestingly, the variations of stretching (υ(OH), 2800–3800 cm⁻¹) and bending (δ(H₂O), 1634 cm⁻¹) vibrations of water molecule became sizeable after the reduction scan. Such a change could be rationalized by the variation of the solvent environment associated with a disruption of the established water molecule network with the quinone molecules through hydrogen bonding[28], as similarly reported for the "water-in-salt" and "molecular crowding" based electrolytes[29,30].

To verify the origin of such a change, the FTIR spectra of 2,6-DHAQ was continuously recorded upon reduction followed by a resting process (350 s). As shown in Supplementary Fig. 9, no change in transmittance at both 1634 and 3250 cm⁻¹ was observed during the entire reduction process, until a sharp increase appeared after the electrolyte was rested for a while, suggesting a subsequent slow chemical process rather than an electrochemical one which disrupted the water molecule network. The FTIR spectrum of the electrolyte at the end of resting is broadly consistent with that of 2,6-DHAQ²⁻, showing it remains the predominant compound after the resting process. In addition, the enchantment of the peaks at 1082 and 1550 cm⁻¹ indicates the presence of 2,6-DHAQ, while the new peaks at around 1105 and 1242 cm⁻¹ are attributed to the presence of 2,6-dihydroxyanthrone (2,6-DHA), suggesting the disproportionation reaction during the resting process producing 2,6-DHAQ and 2,6-DHA. (Supplementary Figs. 10, 11). These results, along with attenuated reduction current in the subsequent CV scan (Supplementary Fig. 12), evince a destructive effect of the changes of surrounding water network, which induces degradation of 2,6-DHAQ during the reduction process. This is in general consistent with previous research works reported in the literature[21].

To unravel the underlying mechanism of the evolving interactions between 2,6-DHAQ and water molecules, density functional theory (DFT) calculations were carried out with Gaussian 16. As shown in Fig. 1b, Supplementary Table 2 and Supplementary Note 1, the adsorption energy of one water molecule onto a 2,6-DHAQ²⁻ is −0.31 eV, indicating a spontaneous uptake process. The presence of hydrogen bond between 2,6-DHAQ²⁻ and H₂O prevents the nucleophilic addition and stabilizes the 2,6-DHAQ²⁻[31]. There are two possible routes in the subsequent reaction process: adsorption of another 2,6-DHAQ²⁻ (2:1 complex) or H₂O molecule (1:2 complex) and the corresponding adsorption energy is 0.14 eV and −0.30 eV, respectively, indicating the uptake of another H₂O is favored. After forming the 1:2 complex, further addition of 2,6-DHAQ²⁻ becomes the rate determining step (ΔE = 0.15 eV), followed by a spontaneous disproportionation reaction. Here, the small energy barrier could be caused by the coulombic repulsion force for the addition of another reduced 2,6-DHAQ. For a deeper insight into the hydrogen bond-promoted formation of the 2:2 complex, electrostatic potential (ESP) maps were plotted. As shown in Supplementary Fig. 13a, there isn't an overlap of ESP maps of the two individual reduced 2,6-DHAQ. However, polarized water molecule would interact with the electron rich oxido group (−O⁻) of the two reduced 2,6-DHAQ by intermolecular electrostatic interactions forming hydrogen bonds[31]. The purple region in Supplementary Fig. 13b represents the hydrogen bond interaction between the H₂O molecule and reduced 2,6-DHAQ. Thus, the coulombic repulsion force between the two reduced 2,6-DHAQ could be overcome by forming hydrogen bonds with H₂O molecule in between. Therefore, the presence of 2,6-DHAQ²⁻ is either in the form of 2,6-DHAQ²⁻·H₂O or 2,6-DHAQ²⁻·2H₂O or 2(2,6-DHAQ²⁻)·2H₂O, or mixture[28]. While the first two could stay stable, the third complex can undergo a further disproportionation reaction and degrade to anthrone[19,21], With the above, the stability issue could intuitively be addressed by preventing the formation of 2(2,6-DHAQ²⁻)·2H₂O, which could be accomplished by molecular engineering of the position of hydroxyl groups.

With the above analysis, the energy change (ΔE) by the formation of 2(DHAQ²⁻)·2H₂O was calculated to select the most stable DHAQ. As shown in Supplementary Table 2 and Supplementary Note 1, the ΔE for 1,2-DHAQ, 1,4-DHAQ and 1,8-DHAQ were calculated to be −0.49, −0.46, and −0.46 eV, respectively, suggesting a spontaneous adsorption process forming 2(DHAQ²⁻)·2H₂O as that of 2,6-DHAQ. In contrast, the ΔE for 1,5-DHAQ is as high as 16 eV, indicating the formation of 2(1,5-DHAQ²⁻)·2H₂O is unfavorable. Thus 1,5-DHAQ is predicted to show chemical stability in AORFB application and is further studied.

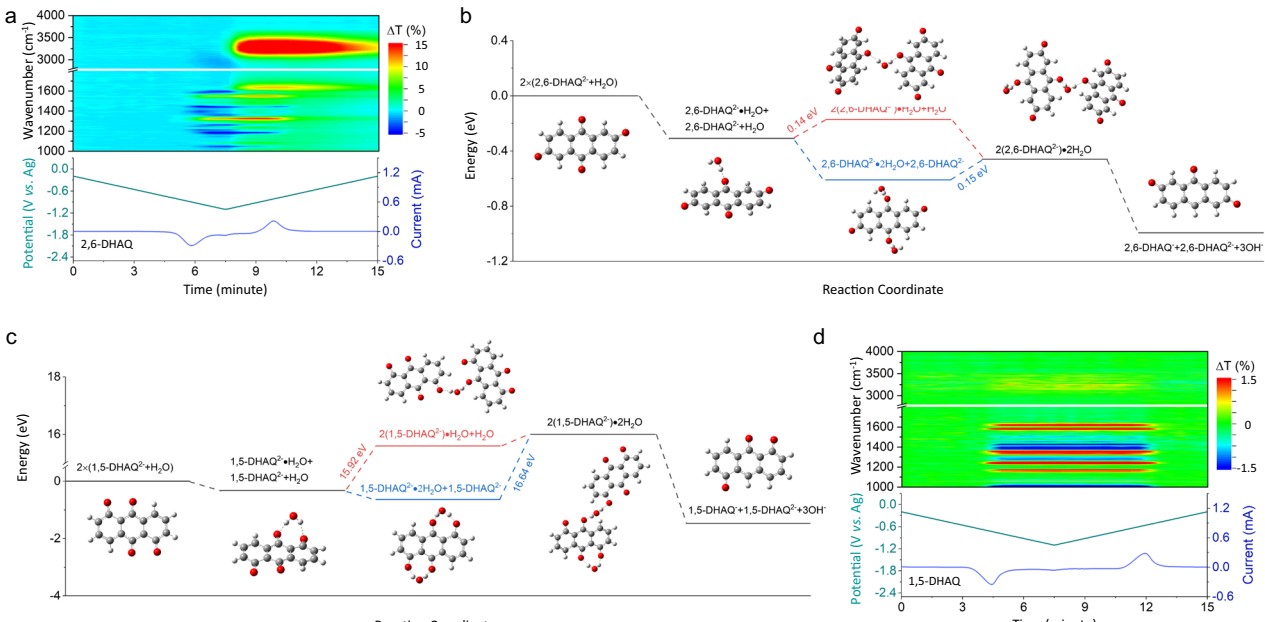

**Fig. 1 | Degradation analysis of DHAQs. a** Evolution of the operando FTIR spectra of 2,6-DHAQ during reduction and oxidation processes in the first cycle of CV scan. **b**, **c** The calculated energy changes of 2,6-DHAQ (**b**) and 1,5-DHAQ (**c**) and water at different redox states. Inserts shows the corresponding optimized structures. **d** Evolution of the operando FTIR spectra of 1,5-DHAQ during reduction and oxidation processes in the first cycle of CV scan. For the FTIR spectra, ΔT is the transmittance difference of the sample relative to that before CV scan. The electrolyte was 0.1 M 2,6-DHAQ or 1,5-DHAQ in 1 M KOH. The details of CV measurement can be found in the method.

The reduction state of 1,5-DHAQ was studied by DFT calculations. As shown in Fig. 1c, Supplementary Table 2 and Supplementary Note 1, the adsorption energy of one water molecule onto a 1,5-DHAQ$^{2-}$ is −0.33 eV, indicating a spontaneous process as that of 2,6-DHAQ. The two hydrogen bonds in 1,5-DHAQ$^{2-}$·H$_2$O (1.58 and 1.90 Å) lower the total energy of the complex despite the strength of each hydrogen bond is weaker than that in 2,6-DHAQ$^{2-}$·H$_2$O (1.56 Å) (Supplementary Fig. 14). The stronger hydrogen bond between 2,6-DHAQ$^{2-}$ and H$_2$O compared with that of 1,5-DHAQ$^{2-}$ leads to a weaker υ(OH) band of H$_2$O molecule, resulting in a larger red shift as confirmed by the FTIR spectra in Supplementary Fig. 7 (υ(OH) band ~2992 vs. ~3258 cm$^{-1}$ for 2,6-DHAQ$^{2-}$·H$_2$O and 1,5-DHAQ$^{2-}$·H$_2$O). Meanwhile, different from 2,6-DHAQ$^{2-}$, it is energetically unfavorable to bring two 1,5-DHAQ$^{2-}$ together by forming hydrogen bonds with a H$_2$O molecule due to a large energy barrier (15.92 eV for the direct uptake of another 1,5-DHAQ$^{2-}$ on 1,5-DHAQ$^{2-}$·H$_2$O and 16.64 eV for the uptake of another 1,5-DHAQ$^{2-}$ on 1,5-DHAQ$^{2-}$·2H$_2$O). Therefore, 1,5-DHAQ$^{2-}$ would preferably form 1,5-DHAQ$^{2-}$·H$_2$O, for which the neighboring oxygen sites are bonded with the same H$_2$O molecule preventing it from joining another DHAQ molecule. Moreover, the protonation of 1,5-DHAQ$^{2-}$ in 1 M KOH (pH = 14) has a positive ΔG of >0.81 eV, indicating the molecule is deprotonated at such a high pH (Supplementary Table 3, Supplementary Fig. 15 and Supplementary Note 2). As a result, 1,5-DHAQ would stay stable upon reduction. All the optimized structures of DHAQH$^-$, DHAQ$^{2-}$, DHAQ$^{2-}$·nH$_2$O and 2-DHAQ$^{2-}$·2H$_2$O can be found in Supplementary Fig. 16. Operando ATR-FTIR was further performed to confirm the reactions of 1,5-DHAQ. The vibrations of C=O and C=C bonds at 1620, 1581 cm$^{-1}$ gradually diminished in the cathodic scans and reappeared in the anodic scans while C−O bond at 1376 cm$^{-1}$ showed a reversed changes[23,32] (Fig. 1d, Supplementary Figs. 7, 17), indicating a good reversibility of 1,5-DHAQ. Not surprisingly, drastically different from 2,6-DHAQ, the stretching (υ(OH), 2800–3800 cm$^{-1}$) and bending (δ(H$_2$O), 1634 cm$^{-1}$) vibrations of water molecule remain stable during the whole electrochemical process and the subsequent resting process (Supplementary Fig. 18), which suggests a good chemical stability of 1,5-DHAQ as predicted by DFT

calculations. This is also corroborated by the nearly unchanged peak currents of the CVs (Supplementary Fig. 19) during the FTIR measurement.

Based on this analysis, the mechanism of hydrogen bonding involved degradation and protection of DHAQ$^{2-}$ was proposed. For 2,6-DHAQ, the hydrogen bonds between two 2,6-DHAQ$^{2-}$ presumably bridge the electron transfer and promote the disproportionation process, eventually forming dimers and resulting in a capacity decay (Fig. 2a). For 1,5-DHAQ, the reduced state (1,5-DHAQ$^{2-}$) would preferably form 1,5-DHAQ$^{2-}$·H$_2$O, for which the neighboring oxygen sites are bonded with the same H$_2$O molecule preventing it from joining another DHAQ molecule (Fig. 2b).

Galvanostatic charge/discharge measurements were conducted to examine the stability of 1,5-DHAQ-based anolytes and its comparison with 2,6-DHAQ. The stability of 1,2-DHAQ-based anolyte was also tested while the solubility of 1,8-DHAQ and 1,4-DHAQ is too low (<10 mM) and they were not tested. As illustrated in Fig. 3a, a standard RFB design was employed to perform the cycling tests, for which DHAQs in 1 M KOH were used as anolyte and ferrocyanide/ferricyanide ([Fe(CN)$_6$]$^{3-/4-}$) as catholyte. The DHAQ|[Fe(CN)$_6$]$^{3-/4-}$ full cells were cycled between 0.4 and 1.8 V at a constant current density of 20 mA/cm$^2$ for around 3 days (70.8 h, 67.9 h and 68.9 h for 1,5-DHAQ, 2,6-DHAQ and 1,2-DHAQ, respectively). The capacity retentions were 78.8%, 65.8% and 96.0% for 1,2-DHAQ, 2,6-DHAQ and 1,5-DHAQ, corresponding to a temporal fading rate of 0.30%, 0.50%, and 0.06% per hour, respectively (Fig. 3b and Supplementary Fig. 20), which confirms the accuracy of the proposed hydrogen bond-mediated protection/degradation mechanism. CV and nuclear magnetic resonance (NMR) measurements analyses were performed to further probe the chemical stability of DHAQs. As shown in Fig. 3c, a new pair of peaks appear at higher potentials (−0.28 V vs. SHE) for 2,6-DHAQ after cycling and the pristine peaks (−0.70 V vs. SHE) become attenuated, suggesting the formation of other species[21]. In comparison, the CV of 1,5-DHAQ remains nearly unchanged. Moreover, the voltage profiles of the last cycle (86th cycle for 2,6-DHAQ and 73rd cycle for 1,5-DHAQ) were plotted. As shown in Supplementary Fig. 21, 2,6-DHAQ showed an

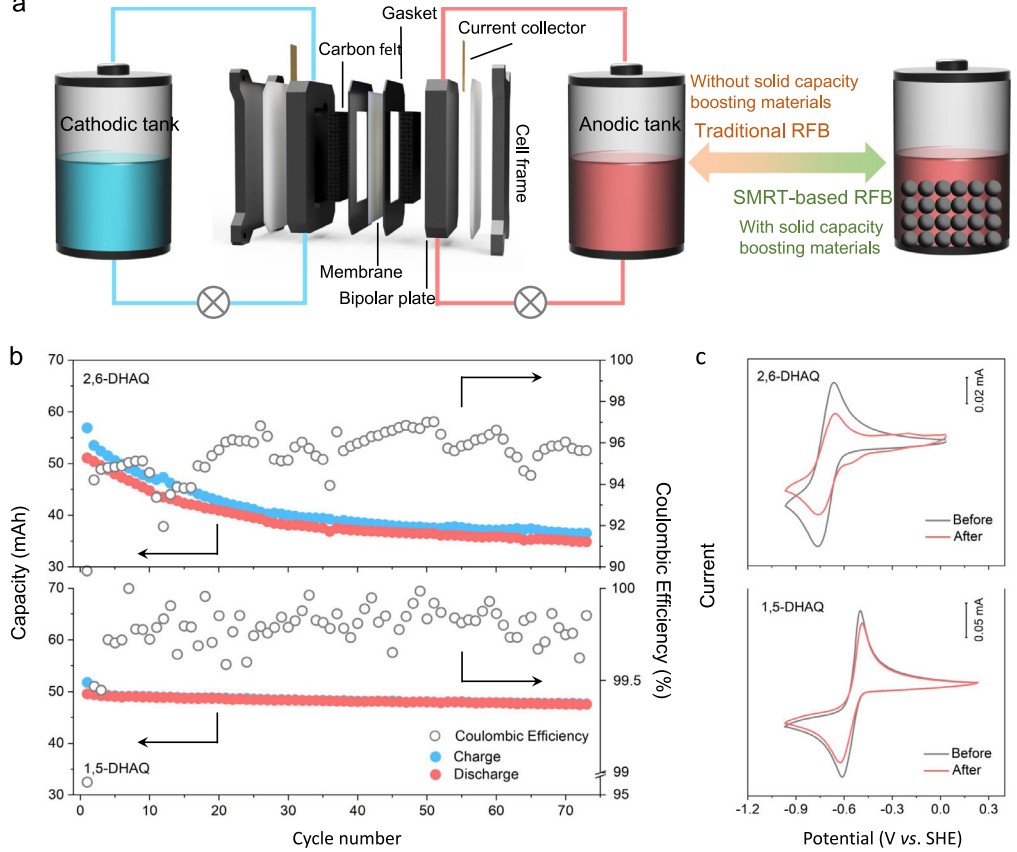

**Fig. 2 | Proposed mechanism of hydrogen bonding involved degradation and protection of DHAQ²⁻. a** The hydrogen bond-mediated degradation process of 2,6-DHAQ²⁻. **b** The hydrogen bond-mediated protection process of 1,5-DHAQ²⁻.

**Fig. 3 | Setup of the batteries and electrochemical performances of conventional RFBs. a** Schematic illustration of the configuration of a conventional RFB with liquid electrolytes filled in both tanks and that of a SMRT-based RFB with solid capacity-boosting material loaded in the anodic tank. The electrode active area was 5 cm². **b** Coulombic efficiency and capacity retention for the anolyte-limited DHAQ|

$[Fe(CN)_6]^{3-/4-}$ full cells. The catholyte was 80 mL 0.25 M $K_4Fe(CN)_6$ + 0.05 M $K_3Fe(CN)_6$/1 M KOH; the anolyte was 10 mL 0.1 M 2,6-DHAQ or 1,5-DHAQ/1 M KOH. The current density was 20 mA/cm². **c** CVs of 5 mM 2,6-DHAQ and 1,5-DHAQ diluted anolyte before and after cycling. The scan rate was 50 mV/s. All tests were conducted at 25 ± 1 °C.

additional voltage plateau compared to the plateau of 2,6-DHAQ/2,6-DHAQ$^{2-}$, which is consistent with the CV results. The higher voltage plateau of both charging and discharging processes corresponds to 2,6-DHAQ/2,6-DHAQ$^{2-}$ redox process while the lower plateau of both charging and discharging processes could be from the reaction of anthrone. In contrast, 1,5-DHAQ showed only one plateau during the charging and discharging processes, corresponding to the redox process of 1,5-DHAQ/1,5-DHAQ$^{2-}$ couple. The degradation product of 2,6-DHAQ was further analyzed by $^1$H-NMR (Supplementary Fig. 22) and a dimer was detected which is consistent with the literature[21]. In contrast, $^1$H-NMR did not detect any apparent changes for 1,5-DHAQ (Supplementary Fig. 23), indicating it is chemically more robust[19]. 1,2-DHAQ behaved similarly to 2,6-DHAQ in both CV and $^1$H-NMR measurements (Supplementary Figs. 24, 25).

## Molecular engineering strategy for DHAQ molecules capacity improvement

Organic redox molecules frequently suffer from low solubility resulting in a poor EES performances[9–11], for which the single molecule redox targeting (SMRT) reaction provides an elegant means to solve the problem without sacrificing battery performance[33]. The SMRT reaction proceeds between a sole soluble redox species and solid capacity-boosting materials via a closed-loop electrochemical-chemical cycle, which effectively exploits the capacity of solid materials and promotes the battery performance. For SMRT, the most important requirement is the match of potential of the soluble redox-active species and solid materials, and the driving force is originated from the Nernstian potential difference induced by the activity changes of the redox species during charging and discharging[25].

Molecular engineering has been reported in the literature to tune the redox potential[34,35], solubility[13], stability[19], and to even activate the inactive electrons[36] of organic molecules for AORFB. Apart from the aforementioned applications, the molecular engineering strategy is proposed here for facile selection of targeted solid capacity-boosting materials for soluble redox-active species. Supplementary Fig. 26a, b and Supplementary Tables 4, 5 present a comprehensive summary of soluble and solid active species including quinone-, viologen-, (2,2,6,6-Tetramethylpiperidin-1-yl)oxyl (TEMPO)- and phenazine-based materials in terms of E$_{1/2}$ and pH of electrolyte solutions. Soluble quinones possess varying redox potentials (−0.72 to 0.85 V vs. SHE) at full pH range (−0.7 to 15), which is also observed in solid quinone-based materials. More importantly, insoluble quinone-based polymers with the same redox center of their monomer show similar redox potential, which provide a convenient way of designing the solid capacity boosters. For example, the E$_{1/2}$ of 2,5-dihydroxy-1,4- benzoquinone (DHBQ) is 0.40 V (vs. SHE) at pH 1, while the polymer poly(2,5-dihydroxy-1,4-benzoquinone-3,6-methylene) also shows an E$_{1/2}$ of 0.40 V (vs. SHE) at the same pH (Supplementary Fig. 26c)[37]. Not only in acidic condition, but it is also observed in alkaline condition (Supplementary Fig. 26d), where 1,4-DHAQ shares the same E$_{1/2}$ with poly(1,4-anthraquinone)[38]. Apart from quinone-based materials, TEMPO-(Supplementary Fig. 26e) and viologen-based materials also show the same property in neutral condition: N,N,N-2,2,6,6-heptamethylpiperidinyl oxy-4-ammonium chloride (TEMPTMA)[39] and poly(2,2,6,6-tetramethylpiperidinyloxy- 4-yl vinylether)[40] share the same E$_{1/2}$ of 0.95 V (vs. SHE), while bis(3-trimethylammonio)propyl viologen tetrachloride (BTMAP-Vi)[41] has a similar E$_{1/2}$ (−0.36 V vs. SHE) with poly(N-4,4-bipyridinium-N-decamethylene dibromide) (−0.35 V vs. SHE)[42]. Phenazine has poor solubility at around 25 °C, and molecular engineering has been used to enhance their solubility[34,43]. The strategy for designing capacity-boosting material is also applicable to phenazine-based materials as long as the soluble and solid materials share the same redox center. For instance, the E$_{1/2}$ of soluble phenazine-7,8-dihydroxyphenazine-2-carboxylic acid (DHPC)[34] is the same as solid tetrapyridophenazine/graphene composite (−0.88 V vs. SHE)[44]. As a result, the molecular engineering strategy shows great potential for quick pairing of solid capacity booster for AORFB.

Based on previous analysis, 1,5-DHAQ is selected as soluble redox-active species and the corresponding polymer- poly(anthraquinonyl sulfide) (PAQS)[45] is used as the capacity-boosting materials for AORFB application. The kinetics of 1,5-DHAQ was first examined with a glassy carbon rotating disk electrode (Supplementary Fig. 27). The diffusion coefficient of the molecule was determined to be $7.24 \times 10^{-6}$ cm$^2$/s and the rate constant is $1.29 \times 10^{-3}$ cm/s, which is higher than most of the inorganic redox-active species[46]. PAQS-based materials – PAQS, PAQS/CB and PAQS/CNT (30 *wt.*% carbon black (CB) and multiwalled carbon nanotube (CNT) in PAQS polymer), were synthesized (Supplementary Figs. 28–32) which showed an E$_{1/2}$ of −0.587, −0.588, and −0.631 V (vs. SHE, Supplementary Fig. 33), respectively. The E$_{1/2}$ of PAQS/CNT is negatively shifted, presumably attributed to the interactions between PAQS and CNT[45]. The material utilization tests were conducted with a 1,5-DHAQ | [Fe(CN)$_6$]$^{3-/4-}$ full cell (Supplementary Fig. 34). After loading 312 mAh equivalent capacity of PAQS granules ($2 \times 1 \times 0.5$ mm$^3$) into the anodic tank, the discharge capacity increased by 56 mAh at 2 mA/cm$^2$, corresponding to a 18% utilization of PAQS (Supplementary Fig. 35). Such a low utilization, which requires a state-of-charge (SOC) of the electrolyte solution as high as 90% in order to trigger the reaction, is ascribed to the sluggish reduction of PAQS by 1,5-DHAQ$^{2-}$ due to the small driving force of reduction as compared to oxidation (Supplementary Note 3). And the head loss for the large-scale application is discussed in Supplementary Note 4. As shown in Supplementary Fig. 35b, as long as there is sufficient reduced PAQS species, rapid oxidation of PAQS can be achieved at a current as high as 40 mA/cm$^2$ during the charge process, indicating the reduction of PAQS is the rate limiting step. Besides the standard potential difference (ΔE°) between 1,5-DHAQ and PAQS, the surface area and conductivity of PAQS granules also play an important role in material utilization[47]. Compared with PAQS (67 m$^2$/g), PAQS/CB and PAQS/CNT composites possess higher surface area (158 m$^2$/g and 166 m$^2$/g, Supplementary Fig. 36). In addition, the highly dispersed CB and CNT in PAQS (Supplementary Fig. 37) enhance the electronic conductivity and are expected to promote the SMRT reaction. As shown in Supplementary Fig. 38, the capacity utilization of PAQS/CB increased to 28% while it was less than 10% for PAQS/CNT as the potential difference is too large to render an effective SMRT reaction.

1,5-DHAQ-PAQS/CB anolyte system was then employed for long-term stability tests using the SMRT-based RFB design (Fig. 3a). The theoretical volumetric capacity and effective electrolyte concentration of PAQS/CB composite are as high as 271 Ah/L and 5.2 M (Supplementary Fig. 39 and Supplementary Note 5), respectively. When paired with [Fe(CN)$_6$]$^{3-/4-}$ catholyte, the full cell is able to yield a cell voltage of 1.03 V (Fig. 4a). The cell was cycled at a constant current density of 20 mA/cm$^2$ with an average discharge voltage of 0.89 V (Fig. 4b). After loading 350 mAh equivalent capacity of PAQS/CB granules ($2 \times 1 \times 0.5$ mm$^3$) into the anodic tank, the discharge capacity increased from 459 to 573 mAh, corresponding to a 33% capacity utilization of PAQS/CB (Fig. 4b). Taking 50% granule loading ratio and the above material utilization into calculation[23], the actual anodic tank volumetric capacity is 47.3 Ah/L (details of calculation can be found in Supplementary Note 5). An extended cycling test was performed to investigate the robustness of the 1,5-DHAQ-PAQS/CB anolyte system. The cell was cycled for over 1000 h and the capacity retention over 100 cycles was 99% with an average coulombic efficiency greater than 99.9%, corresponding to a capacity fading rate of 0.02% per day or 0.01%/cycle (Fig. 4c), which is substantially better than the one without loading granules in the tank (Fig. 3b). Such a stabilized anolyte system with the presence of PAQS/CB granules in the tank could be rationalized by the continued reactions between the 1,5-DHAQ$^{2-}$ and unreacted PAQS even after the charge process stops, which effectively lowers the concentration of the former and consequently the SOC of the

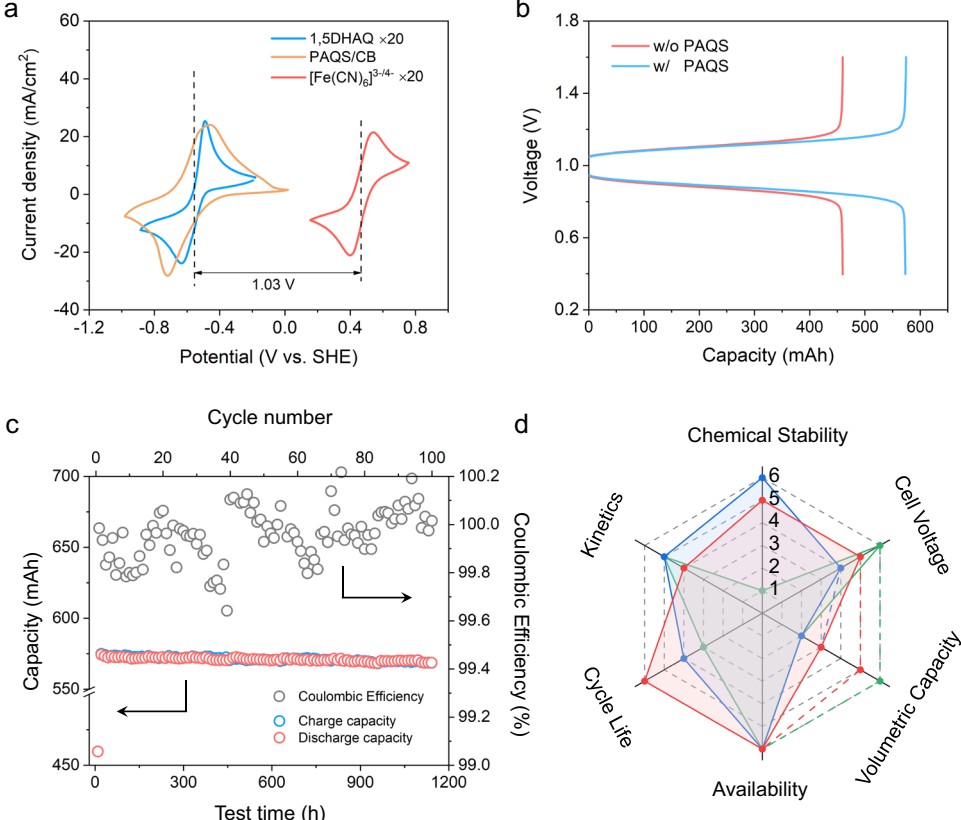

**Fig. 4 | Electrochemical performance of 1,5-DHAQ/PAQS/CB systems and itemized comparison with other quinone-based systems. a** CV curves of 1,5-DHAQ, $[Fe(CN)_6]^{3-/4-}$ and PAQS/CB in 1 M KOH. The scan rate was 50 mV/s. **b** Voltage profiles of 1,5-DHAQ‖$[Fe(CN)_6]^{3-/4-}$ flow cell with (2nd cycle) and without (1st cycle) PAQS/CB granules loaded in the anodic tank at 20 mA/cm². **c** Coulombic efficiency and capacity retention of a flow cell at prolonged cycling test. The catholyte was 250 mL 0.25 M $K_4Fe(CN)_6$ + 0.05 M $K_3Fe(CN)_6$/1 M KOH; the anolyte was 90 mL

0.1 M DHAQ/1 M KOH; 350 mAh equivalent capacity of PAQS/CB granules were loaded into the anodic tank after the first cycle. The current density was 20 mA/cm². **d** Itemized comparison of DPivOHAQ (blue), DHBQ (green) and 1,5-DHAQ-PAQS/CB (red) anolyte system. The dash lines indicate the theoretical volumetric capacity. The detailed information can be found in Supplementary Table 6. All tests were conducted at 25 ± 1 °C.

electrolyte solution. In contrast, most 1,5-DHAQ would be reduced to 1,5-DHAQ²⁻ in conventional RFB (without capacity-boosting materials in the tanks) at the end of charge process because of the high depth of charging process, at which some undesired reactions would be triggered at such a high concentration (0.1 M) of reactive DHAQ species. Hence, the PAQS granules in the tank also act as a SOC buffer, which adjusts the concentrations of 1,5-DHAQ⁰/²⁻ to a moderate level and thus stabilizes the anolyte system.

To investigate the SMRT reactions between 1,5-DHAQ and PAQS/CB, the bonding information of PAQS during repeated CV scans were monitored by operando FTIR (see Supplementary Fig. 40a for the setup). With PAQS/CB coated on the ATR window as a reactive substrate, time-dependent evolution of the FTIR spectra during the SMRT-based reduction and oxidation processes was recorded (Supplementary Fig. 40b and Supplementary Note 6). The stretching vibration of C−O bond of reduced PAQS at 1371 cm⁻¹ gradually appeared during the reduction process of 1,5-DHAQ and vanished with the oxidation of 1,5-DHAQ²⁻. In addition, the reversible changes of vibrations at 1303 and 1262 cm⁻¹ attributed to the aromatic ring vibrations between the neutral and ionized forms (PAQS²⁻)[48], also attest the robustness of PAQS during the SMRT reactions.

Figure 4d summarizes some important metrics of DPivOHAQ, DHBQ and 1,5-DHAQ-PAQS/CB anolyte systems for AORFB applications (see details in Supplementary Table 7). Note DHBQ has a high theoretical volumetric capacity of 231 Ah/L, while the demonstrated one was only 23.2 Ah/L and showed a fast capacity fading. In addition, DPivOHAQ has a low-capacity fading rate of 0.0018%/day at a

demonstrated volumetric capacity of 25.6 Ah/L, while it suffers from a low theoretical volumetric capacity (37.5 Ah/L). Some organic redox mediator-based (quinone, aza-aromatics, and viologen) anolyte systems reported in literature are compiled in the aspects of the achieved volumetric capacity, cell voltage and capacity retention rate (Supplementary Fig. 41 and Supplementary Table 7). Among these materials, the 1,5-DHAQ-PAQS anolyte system presents excellent overall performance for its robustness upon prolonged battery test and higher volumetric capacity with the assistance of redox-targeting reactions. Note that the theoretical volumetric capacity of the SMRT anolyte system could be as high as 139.5 Ah/L at 50% material loading ratio and 100% capacity utilization PAQS/CB granules (details of calculation can be found in the Supplementary Note 5). There is ample room to enhance the capacity by promoting the material utilization. This could be realized by extensive engineering studies, including the optimization of microstructures and surface wettability of the granules accessible to the redox electrolyte, loading/packing of the granules, flow rate and electrolyte conditions, etc.

## Discussion

We have demonstrated the development of a stable anthraquinone-based anolyte system for AORFB based on mechanistic insights gained from operando spectroelectrochemical studies of the reactions and subsequent molecular engineering. With combined operando ATR-FTIR and computational studies, the degradation mechanism of DHAQs were disclosed to be correlated with the formation and disruption of hydrogen bonding network between the reduced quinone

molecules and water, which promotes the intermolecular electron transfer and disproportionation reaction of DHAQ and irreversibly results in the formation of dimer. These findings lead to an effective approach of circumventing the stability issue of DHAQ, by preventing the hydrogen bonding via molecular engineering of the position of hydroxyl groups. 1,5-DHAQ was discovered to present a distinct hydrogen bonding structure which stabilizes the molecule upon reduction. The application of molecule engineering strategy was further extended to guide the designation of capacity-boosting materials. PAQS/CB was used as capacity booster for 1,5-DHAQ-based anolyte system, which showed a volumetric capacity of 47.3 Ah/L (up to 139.5 Ah/L) with a capacity fading rate of 0.02% per day for over 1000 h at 20 mA/cm$^2$ and 25 ± 1 °C.

This study reveals the influence of structural changes of a molecule on its electrochemical properties, which we believe will have interesting implications to other organic redox molecules. In addition, the 1,5-DHAQ-based anolyte in conjunction with PAQS which bears similar redox-active group, opens up an intriguing avenue for the development of solid capacity-boosting material.

## Methods

### Materials
1,5-dihydroxyanthraquinone was purchased from TCI Chemicals. All other chemicals were purchased from Sigma-Aldrich and used without further purification. Nafion 115 (Dupont) membrane was purchased from Chemours. The purity of the following chemicals: 2,6-DHAQ (>90%), 1,5-DHAQ (>85%), 1,4-DHAQ (>96%), 1,8-DHAQ (>96%), and 1,2-DHAQ (>97%).

### PAQS-based materials synthesis
The PAQS polymer was synthesized by following the method reported in the literature[45,49]. To a mixture of 1, 5-dichloroanthraquinone (20.0 g, 72.1 mmol) and sodium sulfide nonahydrate (17.3 g, 72.1 mmol) was added N-Methyl-2-pyrrolidone (180.0 mL) at 25 ± 1 °C. The resulting dark blue color reaction mixture was heated at 200 °C using a paraffin oil bath for overnight under nitrogen atmosphere. During this time, the reaction mixture was turned to dark brown color. The reaction mixture was cooled down to 25 ± 1 °C and filtered the solid by vacuum filtration using Buchner funnel fitted with Whatman filter paper (thickness: 0.18 mm, pore size: 11 μm). The brown color solid was washed with hot (40 ± 1 °C) deionized water (5 × 75 mL) and acetone (>99%, Fisher Scientific) (5 × 75 mL) till the filtrate became colorless. The solid was air dried for 2 h, followed by vacuum dried at 120 °C for overnight to obtain PAQS polymer as a reddish-brown solid (14.4 g, 85%).

The PAQS/CB and PAQS/CNT polymers were also synthesized by following the synthetic procedure described for the PAQS polymer by adding 30% wt./wt. carbon black for PAQS/CB polymer and 30% carbon nanotubes for the PAQS/CNT polymer.

### Granule preparation
PAQS or PAQS/CB or PAQS/CNT, carbon black, Polytetra-fluoroethylene (PTFE) and Na$_2$SO$_4$ (80:15:5:5, w/w/w/w) were mixed in DI water by grinding to form a flake and done by hand using a mortar, which was dried at 60 °C in a vacuum oven for 12 h. PAQS, PAQS/CB and PAQS/CNT granules were then obtained (2 × 1 × 0.5 mm$^3$ in flake shape). The granules were soaked in DI water for 3 days with DI water replacement (in order to dissolve the Na$_2$SO$_4$ in the granules and produce more pores/surface area for redox targeting reaction) and then dry at 60 °C in a vacuum oven for 12 h before use.

### Electrochemical measurements
Cyclic voltammetric (CV) measurements were carried out with an Autolab electrochemical workstation (Metrohm, PGSTAT30) using a three-electrode glass cell composed of a glassy carbon working electrode (diameter: 3 mm), a platinum plate counter electrode (1 × 1 cm$^2$) and an Hg/HgO reference electrode. The glassy carbon working electrode was polished with 0.3 and 0.05 μm of alumina slurry for 2 min and then sonicated in deionized water before every test (power applied during the sonication step of the electrode cleaning is 144 W). The electrolytes used were 10 mL. The dilution process was conducted by adding 0.5 mL 0.1 M DHAQ-based electrolytes into 9.5 mL 1 M KOH solution. The galvanostatic measurement was performed with an Arbin battery tester. Rotatory disk electrode experiments were conducted using a Pine Instruments Modulated Speed Rotator AFMSRCE equipped with a 5 mm diameter glassy carbon working electrode, a Hg/HgO reference electrode, and a Pt counter electrode (1 × 1 cm$^2$). The diffusion coefficient of the oxidized form of 1,5-DHAQ was calculated using the Levich equation (Supplementary Fig. 27b):$i_p = 0.62nFAD^{2/3}v^{-1/6}\omega^{1/2}C$, which relates the mass-transport-limited current to the number of electrons transferred (n), the area of the electrode (A), and the concentration of redox-active species in the electrolyte (C) by plotting the mass-transport-limited current against the square root of the rotation rate with the following parameters: $n = 2$, F = 96485 C/mol, A = 0.196 cm$^2$, C = 1 mM, kinematic viscosity of 1 M KOH = 1.08 × 10$^{-6}$ m$^2$/s. All experiments were conducted at 25 ± 1 °C without using climatic/environmental chamber.

### Flow battery assembly
In the flow cell, Nafion 115 membrane was used to separate the catholyte and anolyte, which was soaked in 1 M KOH solution overnight before use at 25 ± 1 °C. Carbon felt (SGL Carbon; thickness:6 mm; open porosity: 94%) was served as the current collector. Two types of cell structures were used with the active area of 5 cm$^2$ and 13.5 cm$^2$ (Fig. 3a and Supplementary Fig. 34) for battery tests. The one with active area of 5 cm$^2$ was used for long-term tests at a current density of 20 mA/cm$^2$ and the other one with active area of 13.5 cm$^2$ was used for tests with a charging current density of 2 mA/cm$^2$ (and different discharging currents from 2 mA/cm$^2$ to 40 mA/cm$^2$). For DHAQ (including 1,2-DHAQ, 2,6-DHAQ and 1,5-DHAQ)||Fe(CN)$_6$$^{3-/4-}$ cells, 80 mL 0.25 M K$_4$Fe(CN)$_6$ + 0.05 M K$_3$Fe(CN)$_6$/1 M KOH was used as catholyte; 10 mL 0.1 M DHAQ/1 M KOH was used as anolyte. For PAQS (including PAQS, PAQS/CB and PAQS/CNT)/1,5-DHAQ||Fe(CN)$_6$$^{3-/4-}$ cells charging at 2 mA/cm$^2$, 80 mL 0.25 M K$_4$Fe(CN)$_6$ + 0.05 M K$_3$Fe(CN)$_6$/1 M KOH was used as catholyte; 20 mL 0.1 M DHAQ/1 M KOH was used as anolyte. For (PAQS/CB)/1,5-DHAQ||Fe(CN)$_6$$^{3-/4-}$ cell cycling at 20 mA/cm$^2$, 250 mL 0.25 M K$_4$Fe(CN)$_6$ + 0.05 M K$_3$Fe(CN)$_6$/1 M KOH was used as catholyte; 90 mL 0.1 M DHAQ/1 M KOH was used as anolyte. The battery tests were conducted in the N$_2$-filled glovebox at 25 ± 1 °C.

### Characterizations
Operando FTIR measurements were carried out with a PerkinElmer Frontier MIR/FIR system by the attenuated total reflection (ATR) model with a commercialized spectroelectrochemical cell (PerkinElmer Singapore Pte Ltd, the cell consisting a Pt disk (diameter: 3 mm) as working electrode, a Pt plate (4 × 2.5 mm$^2$) as counter electrode and a Ag wire (diameter: 0.5 mm) as reference electrode) to detect the structural evolution of 2,6-DHAQ and 1,5-DHAQ and PAQS/CB composites (see the setup shown in Supplementary Figs. 6, 40) from 4000 to 900 cm$^{-1}$ with a resolution of 4 cm$^{-1}$. The CV scan rate was 2 mV/s, and the working distance was 0.05 mm to the substrate. The blank background was 1 M KOH solution. 0.1 M 2,6-DHAQ and 1,5-DHAQ were dissolved into 1 M KOH to obtain the operando FTIR spectra during CV scans. For SMRT processes between 1,5-DHAQ and PAQS/CB composites. PAQS/CB composites were directly coated on the sampling area, with 0.1 M 1,5-DHAQ in 1 M KOH as the electrolyte. The distance from the working electrode to PAQS/CB is 0.05 mm. Ex situ FTIR measurements for PAQS, PAQS/CB composites and PAQS/CNT composites powder were also carried out with a PerkinElmer Frontier MIR/FIR

system by the attenuated total reflection (ATR) model. All experiments were conducted at 25 ± 1 °C.

Matrix-assisted laser desorption/ionization time-of-flight (MALDI-TOF) mass spectra were recorded on a Bruker Autoflex III spectrometer. The 1H- nuclear magnetic resonance (NMR) analysis of cycled electrolytes was performed by diluting 80 μL of the DHAQ electrolyte in 720 μL of $D_2O$ and spectra were recorded on a BRUKER 500 MHz spectrometer. The cycled electrolytes were collected in the $N_2$-filled glovebox and then sealed before transporting to the NMR test. The solid state 13C-NMR spectra recorded on the AVNEO 400 MHz NMR. The Elemental analysis was done by ThermoFisher Scientific FlashSmart Elemental Analyser. Scanning Electron Microscopy measurements were conducted on Zeiss Sigma 300 and nitrogen adsorption/desorption measurements were conducted on ASAP 2020.

### PAQS polymer FTIR, elemental analysis and MALDI-TOF data
IR (neat) 1674, 1651, 1569, 1306, 1261, 1206, 1129, 975, 809, 753, 704 cm$^{-1}$ (Supplementary Fig. 28); Anal. Calcd. for $Cl(C_{14}H_6O_2S)_5Cl$: C, 68.35; H, 2.46; S, 10.43. Found: C, 67.5; H, 2.48; S, 10.64. MALDI-TOF spectrum indicated that oligomers with up to 20 repeating units were present (Supplementary Fig. 31).

### Computational methods
The structure, energy of DHAQs at different states were calculated by Gaussian 16 program suite, with B3LYP hybrid exchange-correlation functional. Solvation Model Based on Density (SMD) Model was used with water as solvent.

### Reporting summary
Further information on research design is available in the Nature Research Reporting Summary linked to this article.

## Data availability
The authors declare that all data supporting the finding of this study are available within the paper and its Supplementary Information files.

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

## Acknowledgements

This research is supported by the National Research Foundation, Prime Minister's Office, Singapore under its Investigatorship Program (Award No. NRF-NRFI2018-06 (Q.W.)).

## Author contributions

Q.W. and S.H. conceived the study. Q.W. supervised the work. S.H. performed most of the experiments. H.Z. performed the computational studies. S.M synthesized the PAQS-based composites and conducted the NMR tests. J.Z. extensively assisted with battery test and FTIR experiments. Y.Z. performed the BET tests. X.W. advised on granule synthesis. S.H. and Q.W. wrote the paper.

## Competing interests

The authors declare no competing interests.
