## [Peer Review File · Nature Communications]

REVIEWER COMMENTS

Reviewer #1 (Remarks to the Author):

In this article, the authors showed that the stability of a redox flow battery with dihydroxyanthraquinone DHAQ can be improved by a redox targeting strategy. They demonstrated that reduced DHAQ with a OH group on the carbon beta to the carbonyl are less susceptible to degrade to anthrones. They got around the problem of low solubility of DHAQ with OH in position 1, 4, 5, 8 with redox targeting, leading to good volumetric capacities and higher stability. Considering the improved understanding of DHAQs degradation mechanism and results obtained in RFB with the redox targeting strategy, this article could be published in Nature Communications. However, some experimental results are not convincing and according to me the following major (and minors) revisions are necessary:

- **Abstract:** The word "ultra" is not appropriate according to other literature results (Table S17). Ultra-higher volumetric capacity should be replaced by higher and "ultra-stable" should be replaced by stable.
- For easy reading, figures in supplementary materials should appear in the article text in the right order. For example, Fig. S6 is cited line 66 before figures S1-S5.
- The capacity fading in %/cycle should be also given for a better comparison with literature.
- Figure 1a is not clear at all. The spectra should not be superimposed and the values of the applied potentials should be given (not only a color scale). Why transmittance higher than 100% is observed in the spectra of 2,6-DHAQ? There is a problem in recording the spectra. The sample cannot absorb more than the reference.
- Figure 1b: "the sharp increase" of the band at 1634 cm⁻¹ as claimed by the authors lines 86-87 is not visible. The red color is due to a transmittance higher than 120%, which should not be possible. The authors should compare the spectra recorded at the end of the rest time with the spectra of anthrone. It would be useful to conclude on the degradation during the rest period. The same experiment (with a rest time) should be performed with 1,5-DHAQ to check if the n(OH) band is modified or not.
- Figure S12: "the attenuated reduction current in the subsequent CV scan" (line 89-90) is not visible in the figure. It seems to be as stable as in Figure S13.
- It has been reported that anthrones can be reoxidized during discharge. The authors should check if the reoxidation potential of anthrones for 2,6-DHAQ and 1,5-DHAQ are similar. The higher stability of 1,5-DHAQ could be due to a reoxidation process occurring at a less negative potential than 2,6-DHAQ. The potential value have to be given when a "100% depth of discharge" (line 148) is used for each compound.
- Lines 233-235: for a better comparison of PAQS/CB and PAQS/CNT the cyclic voltammograms should be superimposed.
- Figure 4c: the authors should give the correspondence between the number of cycles and the test time.
- Lines 269-270: the peak at 1371 cm⁻¹ is red at the beginning of the reduction and becomes green at the end. So according to the color scale, the transmittance decreases and not increases. Results obtained in Figures S29 and S13 should be compared since the same compound is observed.
- Figure 5b: DHBQ is not on the figure whereas it is discussed in the article text lines 281-282. Demonstrated volumetric capacity would be expected instead of the theoretical value since the capacity retention rate is obtained experimentally. The capacity retention rate cannot be decorrelated from the experimental value of the volumetric capacity.
- Lines 319-321: the purity of the chemicals is an important factor for the solubility and

capacity measurements that has to be given.

- Table S1 has to be revised (example: DHBQ appears several times)
- SI: Line 140 After centrifugation, did the authors filtrate the solution before dilution as it is usually performed?
- Figure S21:a) what is the small reduction wave that appears for high rotation rates? It should be explained d) Lower $E-E^{\circ}$ values should be added to obtain a more accurate value of k_0 .
- Figure S24:a) what are 10, 5 and 2? d) the authors should clearly explain the calculation they made for IR correction.
- Figure S13: since the equivalent capacity are not the same for PAQS/CB and PAQS/CNT, it is difficult to compare.
- The authors never mentioned a possible head loss when a solid is added in the tanks. They should discuss this point.

Reviewer #2 (Remarks to the Author):

Huang et al. propose a more detailed mechanism for bimolecular degradation of hydroxyanthraquinones in redox-flow batteries than currently exists, and examine the performance of 1,5-dihydroxyanthraquinone as a redox mediator for "redox-polymer-boosted" flow batteries. The work is intriguing, as consideration of solvation effects on active reactant degradation has not been previously considered, but the proposed mechanism (this current version, at least) lacks support in some respects, and the performance of the polymer-boosted battery is too meager in comparison to its theoretical potential and previous reports to fully convince me that publication in Nature Communications is warranted. More detailed critique that ought to be addressed follows below:

1. The authors' main conclusion is that strong hydrogen bonding between pairs of reduced hydroxyanthraquinone promotes the irreversible dimerization and capacity loss previously reported by the Aziz group. Although this is plausible, the supporting data and arguments raise a number of questions/concerns:

a. The authors assume that the energetics of transitioning from one reduced hydroxyanthraquinone bound to one water molecule (a 1:1 complex, so to speak) to two reduced species bound to two water molecules (2:2) correlates with the probability of hydrogen bonding between pairs of reduced species. The addition of the second water molecule, however, is odd. Is it not more reasonable to assume that a second reduced quinone adds to the 1:1 complex, with one bridging water molecule, forming a "2:1" precursor complex? Why were the energetics of this reaction not considered in Table S3 and subsequent discussion?

b. The finding that 1,5-DHAQ is intrinsically more stable than 2,6-DHAQ is intriguing, but whether solvation is the main or controlling factor is far from assured. As noted by the authors in section S6, hydrogen bond-promoted formation of pairs of reduced hydroxyanthraquinone is energetically unfavorable across the board with respect to each corresponding 1:1 complex (but much more so for 1,5-DHAQ vs the other isomers); and a very large Coulombic repulsion force is expected between two species that each possess four negative charges. It seems implausible that hydrogen bonding alone is sufficient to significantly increase the driving force for such a bimolecular reaction. The uncertainty is compounded by which sort of precursor complex is relevant. This raises the question of whether the kinetics/energetics of the subsequent anthrone formation or dimerization steps may be rate-limiting for degradation, rather than the energetics of forming the "bi-hydroxyanthraquinone" complex. Particularly for 1,5-DHAQ, one can imagine that there

might be a significant steric hindrance to anthrone dimerization given the vicinal positioning of each pair of enolates.

2. From Figure S7, it appears that 1,5- and 1,8-DHAQ display a shift in redox potential with pH consistent with some degree of proton-coupled electron transfer between 1M and 6 M [OH-], whereas the redox potential of 2,6-DHAQ is independent of pH. This observation agrees with the authors' suggestion that reduced 1,5-DHAQ forms stable complexes with water, but the prospect that protonation of reduced DHAQ may be occurring to some degree should be considered by the authors, as this was not examined as a possibility in DFT calculations and FTIR analysis.

3. Despite the large theoretical capacity that might be achieved by activating the entire PAQS/CB electrode, only 47.3 Ah/L was accessed in the best-performing cell run, corresponding to about 1.75 M of electrons. In terms of accessed capacity and cell voltage, the performance in Figure 4 is on par with the best reported monomeric quinones, but worse in terms of capacity fade rate (~ 0.02 %/day, 5 – 10 \times higher than the best rates) and accessible current density (20 mA/cm² vs 100 mA/cm²).

4. Given the above point, Figure 5b is highly misleading. It makes much more sense to compare accessed rather than theoretical capacities among the various chemistries presented and, arguably, cell voltages rather than redox potential of the capacity-limiting reactant. The linear scale chosen for capacity retention rate is likewise unhelpful because a temporal fade rate of 0.1 %/day is virtually indistinguishable from 0.001 %/day, although both accrue to very different capacity retentions on annual/decadal timescales.

Minor Points:

1. The differing y-axis scales between FTIR data for 2,6- and 1,5-DHAQ challenges comparison of relative intensities of bending and stretching modes for water.

2. It is unclear how 100% depth of discharge was achieved given that cycling was done galvanostatically.

3. Figure 3 summarizes literature data and is therefore more appropriate for the SI than the main text.

4. Changes in capacity and current efficiency vs time in Figure 2b and 4c are difficult to discern because expanded y-axes are used.

Reviewer #1:

In this article, the authors showed that the stability of a redox flow battery with dihydroxyanthraquinone DHAQ can be improved by a redox targeting strategy. They demonstrated that reduced DHAQ with a OH group on the carbon beta to the carbonyl are less susceptible to degrade to anthrones. They got around the problem of low solubility of DHAQ with OH in position 1, 4, 5, 8 with redox targeting, leading to good volumetric capacities and higher stability. Considering the improved understanding of DHAQs degradation mechanism and results obtained in RFB with the redox targeting strategy, this article could be published in Nature Communications. However, some experimental results are not convincing and according to me the following major (and minors) revisions are necessary.

Reply: We thank the reviewer for the remarks. These comments and suggestions are all valuable and helpful for improving our manuscript. We have carefully considered the comments with detailed responses listed below.

1) Abstract: The word “ultra” is not appropriate according to other literature results (Table S17). Ultra-higher volumetric capacity should be replaced by higher and “ultra-stable” should be replaced by stable.

Reply: Thanks for the suggestion. We have replaced the “ultra-higher” volumetric capacity with “high” volumetric capacity and “ultra-stable” with “stable” in the revised manuscript.

2) For easy reading, figures in supplementary materials should appear in the article text in the right order. For example, Fig. S6 is cited line 66 before figures S1-S5.

Reply: Thanks for the suggestion and we have adjusted the order of all figures for easy reading in the revised supplementary materials.

3) The capacity fading in %/cycle should be also given for a better comparison with literature.

Reply: Thanks for the suggestion. The capacity fading rate is 0.01%/cycle for the 1,5-DHAQ/PAQS cell and it has been included in the revised manuscript. Please note that the testing time here is around 11.4 h for one cycle, while for some other redox flow batteries, although more cycles were reported, the testing time was actually only a few minutes each cycle, which may lead to a lower capacity fading rate when using %/cycle for comparison. For the aqueous organic redox flow batteries, it has been suggested that a capacity fading rate in %/day would be more accurate and relevant for practical applications, as the testing time varies from battery to battery (ref.: *Chem. Rev.* 2020, 120, 6467–6489).

4) Figure 1a is not clear at all. The spectra should not be superimposed and the values of the applied potentials should be given (not only a color scale). Why transmittance higher than 100% is observed in the spectra of 2,6-DHAQ? There is a problem in recording the spectra. The sample cannot absorb more than the reference.

Reply: Thanks for the suggestion. We have revised Figure 1a by including the values of applied potentials in the revised manuscript (Figure 1a and 1b). For transmittance higher than 100% observed for 2,6-DHAQ in the original manuscript, it is because 1 M KOH solution was used as

reference for the transmittance measurement. The changes of water environment in the sample could lead to a big difference from the reference. For instance, reducing/enhancing the vibrations of H₂O molecule will result in an increase/reduction of the transmittance, respectively. To avoid confusion, we have changed “Transmittance (%)” to “ ΔT (%)” in the revised manuscript, to instead indicate the transmittance changes of the sample relative to that before CV scan.

Figure 1. Evolution of the FTIR spectra of a, 2,6-DHAQ and b, 1,5-DHAQ during reduction and oxidation processes in the first cycle of CV scan. ΔT is the transmittance difference of the sample relative to that before CV scan.

5) Figure 1b: “the sharp increase” of the band at 1634 cm^{-1} as claimed by the authors lines 86-87 is not visible. The red color is due to a transmittance higher than 120%, which should not be possible. The authors should compare the spectra recorded at the end of the rest time with the spectra of anthrone. It would be useful to conclude on the degradation during the rest period. The same experiment (with a rest time) should be performed with 1,5-DHAQ to check if the n(OH) band is modified or not.

Reply: Thanks for the good suggestion. The increase of the stretching vibration at 3250 cm^{-1} is so strong that the increase of bending vibration at 1634 cm^{-1} becomes invisible. As suggested, the transmittance of both the stretching and bending vibrations has been plotted vs. time, as shown in Supplementary Figure 9, both of which increased during the resting period. As stated previously, the transmittance higher than 120% is caused by the drastic electrolyte environment changes during the reaction process and the reference used for the measurement. As the vibrations of H₂O molecule are weaker than those in the reference (1 M KOH), the transmittance became higher than 100% as observed. To avoid confusion, we have changed “Transmittance (%)” to “ ΔT (%)” in the revised manuscript, to instead indicate the transmittance changes of the sample relative to that before CV scan.

Supplementary Figure 11 compares the FTIR spectrum of the electrolyte at the end of resting with those of 2,6-DHAQ²⁻, 2,6-DHAQ and 2,6-DHA (2,6-dihydroxyanthrone). The FTIR spectrum of the electrolyte at the end of resting is broadly consistent with that of 2,6-DHAQ²⁻, showing it remains the predominant compound after the resting process despite the disproportionation reaction. In addition, the enhancement of the peaks at 1082 cm^{-1} and 1550 cm^{-1} indicates the presence of small quantity of 2,6-DHAQ, while the new peaks at around 1105 cm^{-1} and 1242 cm^{-1} are attributed to the presence of 2,6-DHA (it was synthesized to prove the presence of the molecule as a product

of the disproportionation reaction), suggesting the disproportionation reaction during the resting process producing 2,6-DHAQ and 2,6-DHA. Supplementary Figure 18 shows the same experiment of 1,5-DHAQ (reduction following a resting process). Clearly there almost isn't a change of the transmittance pertaining to the stretching and bending vibrations of H₂O molecule during the whole resting period. This suggests a stable electrolyte environment without a disruption of H₂O molecule network, and consequently the absence of disproportionation reaction of 1,5-DHAQ.

Supplementary Figure 9. a, Evolution of the FTIR spectra of 2,6-DHAQ during reduction and resting processes. b, The plots of transmittance vs. time of 1634 cm⁻¹ and 3250 cm⁻¹. ΔT is the transmittance difference of the sample relative to that before CV scan.

Supplementary Figure 11. FTIR spectra of the electrolyte at the end of resting, 2,6-DHAQ²⁻, 2,6-DHAQ and 2,6-DHA.

Supplementary Figure 18. a, Evolution of the FTIR spectra of 1,5-DHAQ during reduction and resting processes. b, The plots of transmittance vs. time of 1634 cm⁻¹ and 3250 cm⁻¹. ΔT is the transmittance difference of the sample relative to that before CV scan.

6) Figure S12: “the attenuated reduction current in the subsequent CV scan” (line 89-90) is not visible in the figure. It seems to be as stable as in Figure S13.

Reply: Thanks for pointing out the issue. We have now overlaid the reduction currents of both 2,6-DHAQ and 1,5-DHAQ to have a better comparison, as shown in Supplementary Figure 12 and 19. The reduction peak current of 1,5-DHAQ reveals almost no change during the three consecutive CV cycles at around -0.35 mA, while that of 2,6-DHAQ attenuated from -0.29 to -0.26 mA from the first cycle to the third cycle.

Supplementary Figure 12. Replot of the reduction current of 2,6-DHAQ during the three consecutive CV scans of FTIR testing.

Supplementary Figure 19. Replot of the reduction current of 1,5-DHAQ during the three consecutive CV scans of FTIR testing.

7) It has been reported that anthrones can be reoxidized during discharge. The authors should check if the reoxidation potential of anthrones for 2,6-DHAQ and 1,5-DHAQ are similar. The higher stability of 1,5-DHAQ could be due to a reoxidation process occurring at a less negative potential than 2,6-DHAQ. The potential value have to be given when a “100% depth of discharge” (line 148) is used for each compound.

Reply: Thanks for the suggestion. CV tests were conducted for analytes after cycling to confirm if there were anthrones. As shown in Figure 2c, the redox potential of anthrone (2,6-DHAQ) is at around -0.25 V (vs. SHE), which is consistent with the literature (*J. Am. Chem. Soc.* 2019, 141, 8014–8019). In contrast, for 1,5-DHAQ, there was no other redox peak observed during the whole CV range, suggesting the absence of by-product due to the degradation of the molecule. Moreover, the voltage profiles of the last cycle of DHAQs were plotted. As shown in Supplementary Figure 21, 2,6-DHAQ showed an additional voltage plateau, which is consistent with the CV results. The higher voltage plateau corresponds to 2,6-DHAQ/2,6-DHAQ²⁻ redox process and the lower plateau could be from the reaction of anthrone. In comparison, 1,5-DHAQ showed only one plateau during the entire charge and discharge processes, corresponding to the redox process of 1,5-DHAQ/1,5-DHAQ²⁻ couple. As a result, the higher stability of 1,5-DHAQ should not be related to the reoxidation of anthrone.

As the cycling test was conducted galvanostatically (not by limiting the capacity in terms of the theoretical value), the term “100% depth of discharge” is actually not accurate. In the revised manuscript, we thus change the above by describing the detailed testing conditions: “The DHAQ/[Fe(CN)₆]^{3-/4-} full cells were cycled between 0.4 and 1.8 V at a constant current density of 20 mA/cm² for around 3 days”.

Figure 2. c, CVs of 5 mM 2,6-DHAQ and 1,5-DHAQ diluted analyte before and after cycling. The scan rate was 50 mV/s.

Supplementary Figure 21. Voltage profiles of the last cycle of 1,5-DHAQ and 2,6-DHAQ. The current density was 20 mA/cm².

8) Lines 233-235: for a better comparison of PAQS/CB and PAQS/CNT the cyclic voltammograms should be superimposed.

Reply: Thanks for the suggestion. The cyclic voltammograms of PAQS/CB and PAQS/CNT are now overlaid in one graph, as shown in Supplementary Figure 33.

Supplementary Figure 33. CV curves of PAQS, PAQS/CB and PAQS/CNT solid materials in 1 M KOH solution.

9) Figure 4c: the authors should give the correspondence between the number of cycles and the test time.

Reply: Both the cycle number and time have been plotted in Figure 3c based on the experimental data. As there are small variations in capacity in each cycle, there isn't a simple mathematical relation between these two. However, if we assume the capacity be constant for every cycle, considering the total testing time for 100 cycles is 1142 h, the correspondence between the number of cycles and the test time is: $testing\ time = 11.42 \times cycle\ number$.

10) Lines 269-270: the peak at $1371\ cm^{-1}$ is red at the beginning of the reduction and becomes green at the end. So according to the color scale, the transmittance decreases and not increases. Results obtained in Figures S29 and S13 should be compared since the same compound is observed.

Reply: The stretching vibration of C-O bond ($1371\ cm^{-1}$) of PAQS solid material was recorded during the SMRT reaction. Upon reduction, the transmittance at $1371\ cm^{-1}$ gradually decreases, suggesting an increase of C-O signal, associated with the formation of reduced PAQS. However, the results obtained from Figure S29 is for PAQS solid materials while the results obtained from Figure S13 is for 1,5-DHAQ molecule -- they are different compounds. Since both possess the same quinone structure, we compared the evolution of C-O bond upon reaction.

As shown in Figure 1b, during the reduction process, the transmittance of C-O signal at $1376\ cm^{-1}$ gradually increases, indicating the formation of $1,5-DHAQ^{2-}$ by 1,5-DHAQ reduction. While for the same process of redox-targeting reaction of PAQS, the stretching vibration of C-O bond ($1371\ cm^{-1}$) of PAQS solid material gradually emerges and increases (Supplementary Figure 40b), suggesting a chemical reduction process of PAQS through redox-targeting reaction between $1,5-DHAQ^{2-}$ and PAQS forming $PAQS^{2-}$. For the oxidization process, the C-O signals of both $1,5-DHAQ^{2-}$ and $PAQS^{2-}$ disappear, suggesting the oxidization process of $1,5-DHAQ^{2-}$ on the electrode and chemical oxidization of $PAQS^{2-}$ through the redox-targeting reaction. The above discussion has been included in the revised SI under Supplementary Figure 40.

11) Figure 5b: DHBQ is not on the figure whereas it is discussed in the article text lines 281-282. Demonstrated volumetric capacity would be expected instead of the theoretical value since the capacity retention rate is obtained experimentally. The capacity retention rate cannot be decoupled from the experimental value of the volumetric capacity.

Reply: Thanks for the suggestion. DHBQ is now added into the figure, and we have revised the figure by using the demonstrated volumetric capacity instead of the theoretical value, as shown in Supplementary Figure 41. In addition, radar plot is used to have a better comparison of the DHBQ (it was reported with the highest theoretical capacity), DPivOHAQ (it was reported with the best stability) and 1,5-DHAQ/PAQS based electrolytes in the aspects of demonstrated volumetric capacity, cell voltage, duration, kinetics, stability and availability (Figure 3d) in the revised manuscript.

Supplementary Figure 41. Plot of the demonstrated volumetric capacity of some reported organic redox mediator-based aqueous electrolytes and that of this work, versus capacity retention rate and cell voltage. Blue: quinone-based; green: aza-aromatic reactants-based; grey: viologen-based. The full names of the molecules are listed in the Supplementary Table 1.

Figure 3. d, Itemized comparison of DPivOHAQ (blue), DHBQ (green) and 1,5-DHAQ-PAQS (red) anolyte systems (dash line indicates the theoretical volumetric capacity).

12) Lines 319-321: the purity of the chemicals is an important factor for the solubility and capacity measurements that has to be given.

Reply: Thanks for the suggestion. The purity of the chemicals is: 2,6-DHAQ (>90%), 1,5-DHAQ (>85%), 1,4-DHAQ (>96%), 1,8-DHAQ (>96%) and 1,2-DHAQ (>97%). We have included the information in the revised manuscript.

13) Table S1 has to be revised (example: DHBQ appears several times)

Reply: Thanks for pointing out the error. We have now revised Table S1.

14) SI: Line 140 After centrifugation, did the authors filtrate the solution before dilution as it is usually performed?

Reply: Thanks for the question. The solution was not filtrated after centrifugation in the previous experiment. Following the suggestion, we have repeated the experiment by filtrating the solution after centrifugation and then conducted the UV-Vis experiment. The results were given in Supplementary Figure 3f, and the concentration was calculated to be 0.162 M according to a pre-calibrated absorbance-concentration curve of known concentrations of 1,2-DHAQ. This result is rather close to the previously reported one. We have updated the results in the revised SI.

Supplementary Figure 3f. UV-Vis spectrum of 1,2-DHAQ after diluting by 1000 times.

15) Figure S21:a) what is the small reduction wave that appears for high rotation rates? It should be explained d) Lower $E-E^\circ$ values should be added to obtain a more accurate value of k_0 .

Reply: Thanks for the suggestion. The small reduction wave might be caused by oxygen, despite N_2 bubbling. The dissolved trace amount of oxygen can oxidize the reduced DHAQ which regenerates DHAQ with an EC process, leading to an increase in current (small reduction wave) at around -0.7 V. As such, we agree that lower $E-E^\circ$ values should be used for the calculation of k_0 , which have been included in Supplementary Figure 27 and the k_0 was calculated to be 1.29×10^{-3} cm/s.

Supplementary Figure 27. 1,5-DHAQ reduction kinetics in RDE. a) Linear sweep voltammograms of 1 mM 1,5-DHAQ in 1 M KOH on a glassy carbon electrode at rotation rates between 300 and 2100 rpm. b) Levich plot (limiting current versus square root of rotation rate in rad/s) of 1 mM 1,5-DHAQ in 1 M KOH. The slope yields a diffusion coefficient for the oxidized form of 1,5-DHAQ of 7.24×10^{-6} cm²/s. c) Koutecky-Levich plot (reciprocal current versus inverse square root of rotation rate in rad/s) of 1 mM 1,5-DHAQ in 1 M KOH. d) Fitted Tafel plot of 1 mM 1,5-DHAQ in 1 M KOH. The rate constant is calculated to be 1.29×10^{-3} cm/s.

16) Figure S24:a) what are 10, 5 and 2? d) the authors should clearly explain the calculation they made for IR correction.

Reply: Thanks for the suggestion. 10, 5 and 2 in Figure S24 are the current densities. We have revised the figure by inserting the unit.

The resistance of the battery was calculated by the voltage changes at different current. The voltage drops (immediately after a current is applied) at different current relative to OCV are plotted as shown in Supplementary Figure 35c. The liner relationship between voltage and current in (c) indicates that the ohmic resistance from the cell stack dominates the voltage loss and the slope corresponds to the resistance (R). The voltage was then corrected by adding (discharge voltage) or subtracting (charge voltage) the voltage drop: $V_{IR} = V_{battery} \pm I \times R$.

Supplementary Figure 35. a. Voltage profiles of 0.1 M 1,5-DHAQ/[Fe(CN)₆]^{3-/4-} full cells before and after adding 312 mAh equivalent capacity of PAQS granules at different charge and discharge current density. (b) Voltage profiles of the same battery charged at a current density of 2 mA/cm² and discharged at different current densities from 2 mA/cm² to 40 mA/cm². (c). Plots of voltage drop at different current. (d) Replot of the green voltage profile in (a) with IR correction. The catholyte was 80 mL 0.25 M K₄Fe(CN)₆ + 0.05 M K₃Fe(CN)₆/1M KOH; the anolyte was 20 mL 0.1 M 1,5-DHAQ/1M KOH. The electrode active area was 13.5 cm². The liner relationship between

voltage and current in (c) indicates that the ohmic resistance from cell stack dominates the voltage loss.

17) Figure S13: since the equivalent capacity are not the same for PAQS/CB and PAQS/CNT, it is difficult to compare.

Reply: Thanks for the suggestion. We have now conducted another experiment using the same equivalent capacity of PAQS/CB (98 mAh) and the results are shown in Supplementary Figure 38a. The utilization of the solid materials was 20%. The higher utilization could be a result of longer reaction time between 1,5-DHAQ and PAQS/CB solid materials with higher loading in the tank. More detailed discussion on factors influencing reaction yield can be found in our previous paper (ref.: *Chem 3*, 1036–1049, 2017).

Supplementary Figure 38. (a) Voltage profiles of 0.1 M 1,5-DHAQ/[Fe(CN)₆]^{3-/4-} full cells before and after adding 98 or 217 mAh equivalent capacity of PAQS/CB granules at current density of 2 mA/cm². (b) Voltage profiles of 0.1 M 1,5-DHAQ/[Fe(CN)₆]^{3-/4-} full cells after adding 98 mAh equivalent capacity of PAQS/CNT granules at current density of 2 mA/cm². The catholyte was 80 mL 0.25 M K₄Fe(CN)₆ + 0.05 M K₃Fe(CN)₆/1M KOH; the anolyte was 20 mL 0.1 M 1,5-DHAQ/1M KOH. The electrode active area was 13.5 cm². The utilization of PAQS/CB granules increased from 20% to 28% with more granules loading in the tank, which could be caused by the increased chemical reaction time between 1,5-DHAQ and PAQS/CB granules.

18) The authors never mentioned a possible head loss when a solid is added in the tanks. They should discuss this point.

Reply: Thanks for the suggestion. We share the view that it is a very relevant question for practical application, while it is difficult (probably insignificant) to have quantitative discussion of the head loss with such a small lab device. As those studied in packed bed reactors, the presence of solid granules in the tank would induce a pressure drop of the fluid, which leads to additional energy loss of pump. The pressure drop is related to a few factors of the media in the storage tank, such as the porosity and tortuosity of solid granules, packing (loading) of the solid materials, flow rate, etc., which involve extensive chemical engineering optimizations. We hope we could address this in a larger-scale device in future studies. The above discussion has now been included in the SI under Supplementary Figure 35.

Reviewer #2:

Huang et al. propose a more detailed mechanism for bimolecular degradation of hydroxyanthraquinones in redox-flow batteries than currently exists, and examine the performance of 1,5-dihydroxyanthraquinone as a redox mediator for "redox-polymer-boosted" flow batteries. The work is intriguing, as consideration of solvation effects on active reactant degradation has not been previously considered, but the proposed mechanism (this current version, at least) lacks support in some respects, and the performance of the polymer-boosted battery is too meager in comparison to its theoretical potential and previous reports to fully convince me that publication in Nature Communications is warranted. More detailed critique that ought to be addressed follows below:

Reply: We thank the reviewer for the remarks. These comments and suggestions are all valuable and helpful for improving our manuscript. We have carefully considered the comments with the detailed responses listed below.

1a) The authors assume that the energetics of transitioning from one reduced hydroxyanthraquinone bound to one water molecule (a 1:1 complex, so to speak) to two reduced species bound to two water molecules (2:2) correlates with the probability of hydrogen bonding between pairs of reduced species. The addition of the second water molecule, however, is odd. Is it not more reasonable to assume that a second reduced quinone adds to the 1:1 complex, with one bridging water molecule, forming a "2:1" precursor complex? Why were the energetics of this reaction not considered in Table S3 and subsequent discussion?

Reply: Thanks for the question. After the formation of 1:1 complex (one reduced DHAQ bound to one water molecule), there are two possible reaction routes before the formation of 2:2 complex: further addition of one water molecule (1:2 complex) or one reduced DHAQ (2:1 complex). The corresponding formation energies of these intermediates have been calculated and are discussed together with Comment 1b below.

1b) The finding that 1,5-DHAQ is intrinsically more stable than 2,6-DHAQ is intriguing, but whether solvation is the main or controlling factor is far from assured. As noted by the authors in section S6, hydrogen bond-promoted formation of pairs of reduced hydroxyanthraquinone is energetically unfavorable across the board with respect to each corresponding 1:1 complex (but much more so for 1,5-DHAQ vs the other isomers); and a very large Coulombic repulsion force is expected between two species that each possess four negative charges. It seems implausible that hydrogen bonding alone is sufficient to significantly increase the driving force for such a bimolecular reaction. The uncertainty is compounded by which sort of precursor complex is relevant. This raises the question of whether the kinetics/energetics of the subsequent anthrone formation or dimerization steps may be rate-limiting for degradation, rather than the energetics of forming the "bi-hydroxyanthraquinone" complex. Particularly for 1,5-DHAQ, one can imagine that there might be a significant steric hindrance to anthrone dimerization given the vicinal positioning of each pair of enolates.

Reply: Thanks for the question and good suggestion. To have a clearer understanding of the reaction process, the energy changes during the whole process are calculated, as shown in Figure 1c and 1d.

For 2,6-DHAQ, the adsorption energy of one water molecule onto a 2,6-DHAQ²⁻ is -0.31 eV, indicating a spontaneous uptake process. After that, there are two possible routes: adsorption of another 2,6-DHAQ²⁻ or another H₂O molecule and the corresponding adsorption energy is 0.14 eV and -0.30 eV, respectively, indicating the adsorption of another H₂O is favored. After forming the 1:2 complex, further addition of 2,6-DHAQ²⁻ becomes the rate determining step ($\Delta E=0.15$ eV), followed by a spontaneous disproportionation reaction. Here, the energy barrier could be caused by the coulombic repulsion force for the addition of another reduced 2,6-DHAQ for both routes, while the ΔE is only 0.14/0.15 eV. For a deeper insight into the hydrogen bond-promoted formation process of the 2:2 complex, electrostatic potential (ESP) maps were plotted. As shown in Supplementary Figure 13a, there isn't an overlap of ESP maps of the two individual reduced 2,6-DHAQ. However, polarized water molecule would interact with the electron rich oxido group ($-O^-$) of the two reduced 2,6-DHAQ by intermolecular electrostatic interactions forming hydrogen bonds (ref.: *Journal of Power Sources* 501 (2021) 229984). The purple region in Supplementary Figure 13b represents the hydrogen bond interactions between the H₂O molecule and reduced 2,6-DHAQ. Therefore, the coulombic repulsion force between the two reduced 2,6-DHAQ could be overcome by forming hydrogen bonds with a H₂O molecule in between.

For 1,5-DHAQ, after the formation of 1:1 complex, there are also two reaction routes for the formation of 2:2 complex: adsorption of another 1,5-DHAQ²⁻ or H₂O molecule. The adsorption energy of forming 2:1 complex is unfavourably as large as 15.92 eV. In comparison, the adsorption energy of another H₂O molecule (1:2 complex) is -0.31 eV, indicating a spontaneous process. So the reaction is more likely to take the second route and form the 1:2 complex. However, the energy barrier of combining another 1,5-DHAQ²⁻ to the 1:2 complex is extremely large (16.64 eV), suggesting the formation of 2:2 complex is energetically unfavourable. As a result, different from 2,6-DHAQ²⁻, 1,5-DHAQ²⁻ tends to combine only with one H₂O molecule without further combining with another 1,5-DHAQ²⁻.

In summary, the hydrogen bonding process between two 2,6-DHAQ²⁻ and one H₂O molecule overcomes the coulombic repulsion force between the two 2,6-DHAQ²⁻ and facilitates the formation of 2:2 complex, which eventually leads to the degradation of 2,6-DHAQ by a subsequent disproportionation reaction. In stark contrast, 1,5-DHAQ²⁻ has distinct interactions with H₂O and neighboring molecules, which leads to enhanced stability.

Figure 1. The calculated energy changes of c, 2,6-DHAQ, d, 1,5-DHAQ and water at different redox states. Insert shows the corresponding optimized structures.

Supplementary Figure 13. Surface electrostatic potential maps of a, four individual molecules (two $2,6\text{-DHAQ}^{2-}$ and two H_2O) and b, $2(2,6\text{-DHAQ}^{2-}) \cdot 2\text{H}_2\text{O}$ complex.

2) From Figure S7, it appears that 1,5- and 1,8-DHAQ display a shift in redox potential with pH consistent with some degree of proton-coupled electron transfer between 1M and 6 M $[\text{OH}^-]$, whereas the redox potential of 2,6-DHAQ is independent of pH. This observation agrees with the authors' suggestion that reduced 1,5-DHAQ forms stable complexes with water, but the prospect that protonation of reduced DHAQ may be occurring to some degree should be considered by the authors, as this was not examined as a possibility in DFT calculations and FTIR analysis.

Reply: Thanks for the comment. As suggested, the possibility of protonation of the reduced 1,5-DHAQ is analyzed by DFT calculations (Supplementary Table 4). Considering the symmetric

structure of reduced 1,5-DHAQ, the protonation process may take place at two positions as shown in Supplementary Figure 15. The energy of Structure I is lower than Structure II, suggesting the former is more stable and is thus used for the following analysis. The free energy change of the protonation process of Structure I was calculated based on the following equation in 1 M KOH solution:

The pH of the electrolyte would vary with the different protonation process of 1,5-DHAQ. If there isn't protonation, then the protons from 0.1 M 1,5-DHAQ will all react with OH^- in the electrolyte, resulting in a pH of ~13.9 (without considering the volume change); if the 1,5-DHAQ is totally protonated, then there will be no proton dissolved into the solution and the pH of 1 M KOH remains 14. Thus, the pH could be in the range of 13.9-14 after dissolving 0.1 M 1,5-DHAQ in 1 M KOH. Then the ΔG was calculated with pH corrections (ref.: *Proceedings of the National Academy of Sciences* **115**, 6626-6631 (2018)):

$$\Delta G = G_{1,5\text{-DHAQH}^-} + G_{\text{OH}^-} - G_{1,5\text{-DHAQ}^{2-}} - G_{\text{H}_2\text{O}} + pH \times k_B T \ln 10$$

where k_B is the Boltzmann constant and T is the temperature. This leads to a $\Delta G > 0.81$ eV, indicating the protonation is not energetically favorable. Instead, the reduced 1,5-DHAQ would become deprotonated at such a high pH.

Supplementary Figure 15. Possible protonation structures of reduced 1,5-DHAQ.

Supplementary Table 4: Details of free energy calculations.

Structures	1,5-DHAQ	H ₂ O	1,5-DHAQH ⁻ (Structure I)	1,5-DHAQH ⁻ (Structure II)	OH ⁻
Free energy (Hartree), eV	-838.633179	-76.46899	-839.116161	-839.108663	-75.986383

3) Despite the large theoretical capacity that might be achieved by activating the entire PAQS/CB electrode, only 47.3 Ah/L was accessed in the best-performing cell run, corresponding to about 1.75 M of electrons. In terms of accessed capacity and cell voltage, the performance in Figure 4 is on par with the best reported monomeric quinones, but worse in terms of capacity fade rate (~0.02 %/day, 5 - 10× higher than the best rates) and accessible current density (20 mA/cm² vs 100 mA/cm²).

Reply: Thanks for the comment. We fully agree that there is ample room to improve the materials utilization and cycling stability, by unleashing the full potential of the DHAQ-mediated PAQS

anolyte system. This could be realized by extensive engineering studies, including the optimization of microstructures and surface wettability of the granules accessible to the redox electrolyte, loading/packing of the granules, flow rate and electrolyte conditions, etc. As stated above, this work has a greater focus on the mechanistic understanding on the subtle structure-stability relation, and meanwhile presents a molecular engineering strategy for fast solid capacity boosting materials selection by using redox polymers bearing similar moieties to the soluble redox mediators in the electrolyte. The engineering optimization will be performed in forthcoming studies.

4) Given the above point, Figure 5b is highly misleading. It makes much more sense to compare accessed rather than theoretical capacities among the various chemistries presented and, arguably, cell voltages rather than redox potential of the capacity-limiting reactant. The linear scale chosen for capacity retention rate is likewise unhelpful because a temporal fade rate of 0.1 %/day is virtually indistinguishable from 0.001 %/day, although both accrue to very different capacity retentions on annual/decadal timescales.

Reply: Thanks for the suggestion. Following the suggestion, we have revised the figure by using the demonstrated capacity instead of the theoretical value, and the cell voltage instead of redox potential, as shown in Supplementary Figure 41. More detailed information is summarized in Supplementary Table 7, which could clearly distinguish the difference in capacity fading rate. Moreover, radar plot is used to have a better comparison of the DHBQ (it was reported with the highest theoretical capacity), DPivOHAQ (it was reported with the best stability) and 1,5-DHAQ/PAQS-based electrolytes in the aspects of demonstrated volumetric capacity, cell voltage, duration, kinetics, stability and availability (Figure 3d).

Supplementary Figure 41. Plot of demonstrated volumetric capacity of some reported organic redox mediator-based aqueous anolytes and that of this work, versus capacity retention rate and cell voltage. Blue: quinone-based; green: aza-aromatic reactants-based; grey: viologen-based. The full names of the molecules are listed in the Supplementary Table 1.

Figure 3d, Itemized comparison of DPivOHAQ (blue), DHBQ (green) and 1,5-DHAQ-PAQS (red) anolyte system (dash line indicates the theoretical volumetric capacity).

5) The differing y-axis scales between FTIR data for 2,6- and 1,5-DHAQ challenges comparison of relative intensities of bending and stretching modes for water.

Reply: Thanks for the suggestion. We have adjusted the y-axis scales to the same value for 2,6-DHAQ and 1,5-DHAQ, as shown in Supplementary Figure 7. For better clarity, transmittance difference (ΔT) of the sample relative to that before CV scan is used in place of transmittance relative to the reference solution (1 M KOH), in order to eliminate complexity in data interpretation and avoid confusion.

Supplementary Figure 7. Evolution of the FTIR spectra of 2,6-DHAQ and 1,5-DHAQ during reduction and oxidation processes in the first cycle of CV scan. ΔT is the transmittance difference of the sample relative to that before CV scan.

6) It is unclear how 100% depth of discharge was achieved given that cycling was done galvanostatically.

Reply: Thanks for pointing out the issue. As the cycling test was conducted galvanostatically (not by limiting the capacity to the theoretical value), the term “100% depth of discharge” is actually not accurately used. In the revised manuscript, we thus change the above by describing the detailed testing conditions: “The DHAQ|[Fe(CN)₆]^{3-/4-} full cells were cycled between 0.4 and 1.8 V at a current density of 20 mA/cm² for around 3 days”. “The redox-targeting based 1,5-DHAQ/PAQS |[Fe(CN)₆]^{3-/4-} full cell was cycled between 0.4 and 1.6 V at a current density of 20 mA/cm² for more than 1000 h.

7) Figure 3 summarizes literature data and is therefore more appropriate for the SI than the main text.

Reply: Thanks for the suggestion. Following the suggestion, we have removed the figure from the main text and shifted to SI.

8) Changes in capacity and current efficiency vs time in Figure 2b and 4c are difficult to discern because expanded y-axes are used.

Reply: Thanks for the suggestion. The scale of y-axes has now been modified for both figures in the revised manuscript.

REVIEWER COMMENTS

Reviewer #1 (Remarks to the Author):

I consider that the authors did all the corrections and that the article can be now published in Nature Communications.

Reviewer #2 (Remarks to the Author):

The authors have provided some helpful additional calculations (Figure 1c and d) of the energetics of different pathways to a two-quinone complex starting from reduced 1,5- and 2,6-DHAQ. The results comprise compelling support for their hypothesis that hydrogen bonding is likely rate-determining for anthrone formation. Besides my initial concern about the mediocre performance of the flow cells assembled with PAQS/CB electrodes (they fair no better than the best-performing conventional un-mediated aqueous organic flow cells), the following issues should be addressed if this work is accepted:

1. Based on DFT calculations, the authors reject the possibility that 1,5-DHAQ gets protonated at high pH, but offer no potential explanation for why there is a decrease in 1,5-DHAQ redox potential from 1 M to 6 M KOH (which is consistent with proton-coupled electron transfer).
2. In SI figure 41, A1/L should be Ah/L.
3. Numbers should be added to the axes in the spider plot of Figure 3d.

Reviewer #1:

I consider that the authors did all the corrections and that the article can be now published in Nature Communications.

Reply: We thank the reviewer for the remarks.

Reviewer #2:

The authors have provided some helpful additional calculations (Figure 1c and d) of the energetics of different pathways to a two-quinone complex starting from reduced 1,5- and 2,6-DHAQ. The results comprise compelling support for their hypothesis that hydrogen bonding is likely rate-determining for anthrone formation. Besides my initial concern about the mediocre performance of the flow cells assembled with PAQS/CB electrodes (they fair no better than the best-performing conventional un-mediated aqueous organic flow cells), the following issues should be addressed if this work is accepted:

Reply: We thank the reviewer for the remarks. These comments and suggestions are all helpful for improving our manuscript. We have carefully considered the comments with the detailed responses listed below. For the battery performance of the 1,5-DHAQ mediated PAQS/CB anolyte system, we fully agree that there is ample room to improve the materials utilization and cycling stability by extensive engineering studies, including the optimization of microstructures and surface wettability of the granules for enhanced conductivity of charges and accessibility to the redox electrolyte, loading/packing of the granules, flow rate and electrolyte conditions, etc.

1) Based on DFT calculations, the authors reject the possibility that 1,5-DHAQ gets protonated at high pH, but offer no potential explanation for why there is a decrease in 1,5-DHAQ redox potential from 1 M to 6 M KOH (which is consistent with proton-coupled electron transfer).

Reply: Thanks for the comment. One possible explanation for this negative shift of redox potential could be the activity coefficient changes of the reduced and oxidized species at high pH.

Based on the Nernst equation:

$$E = E^0 - \frac{RT}{nF} \ln \left(\frac{\alpha_{red}}{\alpha_{ox}} \right) = E^0 - \frac{RT}{nF} \ln \left(\frac{\gamma_{red}}{\gamma_{ox}} \right) - \frac{RT}{nF} \ln \left(\frac{C_{red}}{C_{ox}} \right) = E_f^0 - \frac{RT}{nF} \ln \left(\frac{C_{red}}{C_{ox}} \right)$$

Where E^0 is the standard potential, α is the activity of the reduced and oxidized species, γ is the activity coefficient of the reduced and oxidized species, C is the concentration of reduced and oxidized species, and E_f^0 is the formal potential:

$$E_f^0 = E^0 - \frac{RT}{nF} \ln \left(\frac{\gamma_{red}}{\gamma_{ox}} \right)$$

Considering that:

$$E_{1/2} = E_f^0 + \frac{RT}{nF} \ln \left(\frac{D_{red}}{D_{ox}} \right)^{0.5}$$

Assuming $D_{red} = D_{ox}$, then:

$$E_{1/2} = E_f^0 = E^0 - \frac{RT}{nF} \ln \left(\frac{\gamma_{red}}{\gamma_{ox}} \right)$$

As a result, $E_{1/2}$ may vary with the activity coefficients of the reduced and oxidized species. For different pH conditions: 1 M KOH, 3 M KOH and 6 M KOH, the changes of supporting electrolyte concentration or ionic strength presumably have different influence on the activity coefficient of the reduced and oxidized species (interactions with surrounding charged species) as those observed in other systems (References: J. Am. Chem. Soc. 1950, 72, 9, 3918–3920; J. Am. Chem. Soc. 1964, 86, 14, 2790–2792), and thus leads to a different $E_{1/2}$. The above discussion has been included in the revised SI in Section S5.

2) In SI figure 41, Al/L should be Ah/L.

Reply: Thanks for pointing out the error. We have revised the unit in SI part.

3) Numbers should be added to the axes in the spider plot of Figure 3d.

Reply: Thanks for the suggestion. We have revised the spider plot of Figure 3d by including the numbers.

REVIEWERS' COMMENTS

Reviewer #2 (Remarks to the Author):

My concerns have been satisfactorily addressed.